# An Exploration of the Effects of Cross-Modal Tasks on Selective Attention

**DOI:** 10.3390/bs13010051

**Published:** 2023-01-06

**Authors:** Krithika Nambiar, Pranesh Bhargava

**Affiliations:** Birla Institute of Technology & Science, Pilani—Hyderabad Campus, Hyderabad 500078, India

**Keywords:** perceptual load, selective attention, distractor processing, distractor congruency

## Abstract

Successful performance of a task relies on selectively attending to the target, while ignoring distractions. Studies on perceptual load theory (PLT), conducted involving independent tasks with visual and auditory modalities, have shown that if a task is low-load, distractors and the target are both processed. If the task is high-load, distractions are not processed. The current study expands these findings by considering the effect of cross-modality (target and distractor from separate modalities) and congruency (similarity of target and distractor) on selective attention, using a word-identification task. Parameters were analysed, including response time, accuracy rates, congruency of distractions, and subjective report of load. In contrast to past studies on PLT, the results of the current study show that modality (congruency of the distractors) had a significant effect and load had no effect on selective attention. This study demonstrates that subjective measurement of load is important when studying perceptual load and selective attention.

## 1. Introduction

Successful engagement with the environment requires interaction with stimuli presented to the various sensory organs. When performing tasks, we are faced with a continual stream of information in the form of perceptual stimuli. Attention is the mechanism that helps to process various stimuli presented by the different sensory organs. However, attending to stimuli has associated costs, because perceptual processing capacity is an exhaustible resource [1,2]. One way to deal with this is to attend only to relevant information, e.g., information from the relevant stimuli, i.e., the target, while ignoring distracting information. This is achieved through selective attention allowing for the preferential processing of the presented sensory information relating to features, locations, orientation, and modalities [3,4].

A pertinent question in this regard relates to the stage of information processing during which selective attention applies. So-called early-selection theories claim that task-relevant information is selected at an early stage of processing, allowing targets to be perceptually encoded while ignoring distractors [5,6]. On the other hand, so-called late-selection theories claim that both the target and distractors are perceptually encoded in the initial stages of processing. It is only at a later post-perceptual stage that target selection for further processing takes place [7].

With perceptual load theory (PLT), Lavie [8] brought together these two types of theory. PLT posits that perceptual processing at all times involuntarily processes information to its full capacity. While performing a task, top-down identification of relevant and irrelevant information is led by the voluntary control of perception. Through selective attention, the task-relevant aspects of the stimuli are prioritized for processing. If the task is low-load, attending to the target does not engage the whole perceptual process, but as perceptual processing is involuntary and must be used to its full capacity, irrelevant information including distractions is processed along with relevant information relating to the target. However, if the task is high-load, it consumes all the available processing capacity in attending to the target, leaving no spare capacity for processing the distractions [1,9]. Thus, according to PLT, the stage at which selective attention applies and the allocation of processing capacity depend on the load induced by the task at hand. The load induced by a task is dependent on cognitive demand as well as the perceptual properties of the task [9,10,11].

Thus, the concept of high or low perceptual load is operationalized on the basis of distractors and targets [12,13]. The task is considered high perceptual load when the distractions or non-target information are not processed. The task is considered low-load when the attentional resources spill over and process distractions along with the target information, a situation known as distractor interference. Corroborating this, results from many studies on PLT have found that the perceptual demand induced by the task determines the allocation of attentional resources. High-perceptual-load tasks preclude processing irrelevant aspects of the stimuli [2,14,15].

Since perceptual load is observable only by the processing (or lack thereof) of distractions, the study and observation of perceptual load can involve manipulations of aspects of tasks, such as the target itself, task-relevant and task-irrelevant distractors, or the objective of the task. Studies involving PLT use these aspects to manipulate and study perceptual load in three different ways. One of these ways is to alter the number of items displayed during perceptual recognition tasks. Increasing the number of items on display increases the complexity of the task, hence increasing the perceptual load. In one of her early studies, Lavie [16] implemented this method of load manipulation by having the target appear in one out of six possible positions on the visual display, with five positions empty under low-load conditions. For high-load conditions, the five positions were occupied by non-target letters. Another method of load manipulation is by keeping the nature and/or number of displayed conditions unchanged while altering the number of operations to be performed to complete the task. Increasing the number of perceptual operations involved in a task increases the task’s complexity. In one such study, along with manipulating the load by increasing or decreasing the number of letters in a visual search task, the demands on perceptual judgement were varied by comparison with length discrimination or colour detection using identical stimuli [11].

Perceptual load is also affected by increasing the similarity, also called the congruency, between the target and the distractors [17,18,19]. Congruent distractors, which have similar properties to the target, compete with the target for attentional resources. Incongruent distractors, which are dissimilar to the target, do not compete with the target [20]. Studies employing the letter-search paradigm reported that when searching for a target such as the angular letter X, surrounded by congruent distractors like the angular letter Z, the response was faster (shorter response time). If other irrelevant non-target stimuli, e.g., cartoon faces, were also present, these were not even processed [10,15,21]. However, when searching for the letter X among incongruent distractors such as the circular letter O, the response was slower (longer response time), and irrelevant stimuli were also processed. Thus, for congruent distractors, the perceptual similarity between the targets and the distractions led to higher perceptual load, exhausting the attentional resources. In the case of incongruent distractors, the task was less demanding and perceptual load stated as low, leaving spare attentional capacity to process task-irrelevant information [22]. In short, congruency of distractors and target has a significant effect on selective attention but only when the perceptual load of the task is low.

### 1.1. Issues with PLT

There are two main issues affecting studies reporting the results of PLT, namely the circularity problem and the limiting of experiments to the visual domain. The circularity problem refers to the circular characterization of perceptual load. On one hand, distractor interference is assumed to depend on whether the task is high- or low-load; on the other, whether the task is high- or low-load itself depends on whether or not the distractor causes interference [12,13]. Thus, there is no independent validation of whether a task is high- or low-load. The experimenter testing the effect of load on distractor interference decides a priori whether a task is high- or low-load, and accordingly interprets the performance of participants. To address this issue, in the current study, instead of the researchers assuming the extent of the task load, the participants were asked to subjectively rate the load separately after the experimental tasks.

The second issue with PLT is that the knowledge gained in this context about selective attention and perceptual load is based largely on the visual domain, because studies conducted under PLT have been predominantly in the visual domain. However, our experience of the real world is multimodal in nature, i.e., involving more than one modality. To imitate better the real world scenario of selective attention in studies of perceptual load, it is important to study tasks involving both visual and auditory stimuli [23]. Because of the evolutionary difference in the functions of vision and hearing in the real world, a difference may also exist in the way an individual interacts with auditory and visual distractions while performing tasks that need attention [24,25,26].

A small but growing niche of studies have begun to explore the role of auditory modalities in perceptual load, with some reporting unimodal experiments with auditory and visual modalities. These studies found that selective attention is dependent on load, irrespective of the modality. For instance, in a figure–ground segregation study reported by Molloy et al. [27], task-irrelevant sounds were presented during the performance of a visual search task and the results revealed a ‘clear magnetoencephalography neural signature of figure-ground segregation in conditions of low visual load, which was substantially reduced in conditions of high visual load’. Therefore, for both of these modalities, distraction recognition depends on the level of perceptual load. Other studies that conducted unimodal experiments with auditory and visual stimuli included tasks in which the distractions were presented in the same modality as the target [28,29,30,31]. Several studies involved tasks using multiple modalities for targets and distractors, but these did not take the congruency factor into consideration [24,25,32,33].

Thus, the research gap in the literature arises from a dearth of studies that (i) involve cross-modal tasks (target from one modality, distractor from another), while (ii) taking into consideration the congruency of the distractors, and (iii) including subjective measurement of load from the participants. 

### 1.2. Current Study

Studies of multisensory integration have demonstrated that humans perceive their environment better when they are able to bind perceptual information from different senses and combine this information into a coherent representation. Therefore, in order to study cross-modal perceptual congruency, one must use an object that can be perceived simultaneously by the corresponding senses [34,35]. One method employed to achieve this is the use of a picture of an animal (e.g., dog or cat) as the visual stimulus and a corresponding or non-corresponding call (e.g., barking or mewing) as the auditory stimulus [36]. The problem with this is that the buttons for receiving the participant’s response need to be labelled with pictures (e.g., of a dog and a cat), which supplies an over-representation of the visual stimulus (i.e., not only as the visual task stimulus, but also on the button label) compared with the auditory stimulus (because there is no ‘auditory button’). An acceptable solution to this is to label the buttons with words (e.g., DOG and CAT), which requires the participant to read the word on the button. In the current study, we built upon this solution.

In languages with alphabetic writing systems (e.g., English), the textual spelling and phonological pronunciation of a word are integrally connected through orthographic knowledge [37,38]. For unknown and less familiar words, the speaker of a language would read the words piecemeal, but for common and familiar words, the spelling and pronunciation are stored together as a picture–sound unit in the individual’s lexical orthographic knowledge, such that the sight of a printed or written word invokes its pronunciation, and vice-versa. Thus, words contained in the individual’s lexical orthographic knowledge, i.e., very frequent and highly familiar words, undergo cross-modal (visual and auditory) activation. Correspondingly, we assume that the task of reading one word while listening to a different word would represent a cross-modal target–distractor paradigm. With this in mind, we used frequently occurring Indian English words and their corresponding utterances as cross-modal stimuli in the current study. The task was perceptual in nature because it involved integration of two perceptual modalities in the form of targets and distractors. This also allowed the buttons to be labelled with single letters (initial letters of the names of the stimuli) which helped to avoid over-representation of the visual or auditory stimuli. Within this paradigm, because of the use of meaningful words, the semantic congruency was cognitive in nature.

We employed the aforementioned model in the current study design to address the previously mentioned research gap by: (i) incorporating cross-modality in choosing the targets and distractors, i.e., for a visual target, then the distractor was auditory (and vice versa), (ii) using two different types of distractors, i.e., congruent and incongruent, and (iii) asking the participants themselves to rate the load of the tasks after completion. The overarching research objective was to see if congruency and modality of distractors (vis-à-vis the target) affected the perceptual load of tasks.

On the basis of results from previous studies involving PLT, in the present study it was expected that modality would not play a significant role in selective attention; thus:

Expectation 1: There would be no significant differences in the response times and accuracy scores of the participants for tasks from any modality.

Furthermore, results from PLT studies also showed that congruency of distractions has no significant effect on the performance of participants, thus:

Expectation 2: There would be no significant difference in the performance scores of participants for tasks with varying distractor congruency.

If the performance scores of participants in bimodal audio–visual tasks do not fluctuate, we can conclude that the results align with the existing research and study results relating to PLT. Such a result would indicate that selective attention functions in a uniform way irrespective of modality, and that the congruency of distractors on target recognition in a task is load-dependent in its effect rather than modality-dependent. If the results do show differences, it could indicate that selective attention varies with the nature and modality of the task. This would imply that apart from load being induced by the task itself, i.e., some tasks being inherently difficult or easy and thereby classified as high- or low-load tasks, the inclusion of targets and distractors from two different modalities affected selective attention, leading to slower reaction times and lower accuracy scores.

The current research incorporated a post-experiment questionnaire to measure the load of the auditory and visual tasks included in the study. This provided a subjective measurement of load as indicated by the participants. The participants were asked to recall the task they completed involving a particular modality and to rate the task using the parameters stated in the questionnaire. Consequently, two sets of the questionnaire were distributed in order of completion of the experiments. The scores given by the participants were assessed by the experimenters to discover whether the reaction times and accuracy scores across the two tasks with different modalities were indeed affected by the task loads. This post-hoc measurement is considered important to provide an unbiased interpretation of load, which could not be achieved if the load were predetermined by the experimenters. It is important to note that this post-experiment questionnaire provided results that were indicative of the cognitive load or the working memory load, because it involved participants’ recall [1].

## 2. Materials and Method

### 2.1. Participants

Thirty-one participants (14 females; mean age = 30 years), with reported normal hearing and vision, were recruited from BITS Pilani’s Hyderabad campus. The participants received rewards of stationery for their participation. Each participant took part in the two experimental tasks on the same day. The participants provided informed consent before their participation.

### 2.2. Apparatus and Stimuli

Audacity^®^ (version 3.0.0) was employed to record and process the auditory inputs. These inputs were then utilized for construction of the experiment in PsychoPy Experiment Builder version 3.0. The open-source version of RStudio, the integrated development environment (IDE) for R, was employed to analyse the data. RStudio used the statistical tool R (64-bit, version 3.5.1) for the analysis. All the packages that were applied in R were installed through the R-Cran cloud library. For plotting the graphs generated by R, Rcmdr package version 2.5-1 was used. The auditory inputs were delivered using Audio-Technica ATH-M20x over-the-ear headphones. The NASA task load index (TLX) Version 1.0 paper and pencil package was used for subjective task-load ratings [39].

Three words for colours, namely *Red*, *Green*, and *Blue*, and three non-colour words, namely *Pen*, *Lid* and *Mug* were recorded spoken by a female voice in a sound-treated chamber. The words were monosyllabic, commonplace English terms, 500 ms in duration, and normalized in intensity with each other.

### 2.3. Procedure

For experiments 1 and 2, the participants were seated in a sound-treated chamber and presented with visual stimuli on a computer screen and auditory stimuli through headphones. They recorded their responses with mouse clicks. Each of the experiments comprised a training session followed by two experimental tasks. On-screen and verbal instructions from the experimenters were provided to the participants during the training, and before (but not during) each task. The participants kept the headphones on during the training and the tasks.

The experiment commenced with training in which the participant was familiarized with the user interface, the stimuli, and the process. During the training, participants were permitted to adjust the volume of the audio and the brightness of the screen to meet their preference. These settings then remained unchanged for that participant for both experimental tasks. The training was repeated until a participant was confident and had no more questions.

In order to reduce any strategy-based effects of modality on the performance of participants, and to compensate for any potential bias, half of the participants performed the visual task first, followed by the auditory task. The other half completed the auditory task first, followed by the visual task.

#### 2.3.1. Experiment 1: Visual Task (VT)

The effect of selective attention on visual modality was tested using the visual task. In VT, the target of the task was the visual stimulus, and the distractor was the auditory stimulus. Refer to Figure 1 for a representation of a typical trial. Before the task and during the training, the participants were asked to ignore any auditory stimuli they might hear during the task. For the first 500 ms of each trial, participants were presented with a ‘+’ fixation symbol on the screen, along with an alerting auditory tone delivered through the headphones. Soon after the fixation and the alert tone, the visual target and auditory distractor were presented simultaneously on the screen and through the headphones, respectively.

The visual target was a randomly selected word from the pool of only the colour words. This word was displayed on the screen for 500 ms, in black Times New Roman font. The auditory distractor was an auditory stimulus randomly selected from the pool of colour and non-colour words. The visually presented word in the VT was always a colour word, therefore a colour word as an auditory stimulus was a *congruent* distractor, while a non-colour word as an auditory stimulus was an *incongruent* distractor.

After the presentation of the target and distractor, three on-screen buttons appeared with the text ‘R’ for red, ‘B’ for blue, and ‘G’ for green. The task was to identify the colour word presented visually on the screen, by clicking the corresponding on-screen button using the mouse. Immediately after the response from the participant was received through the mouse click on any of the three on-screen buttons, the next trial was presented automatically.

There were 18 unique pairs of target visual stimuli (3 colour words) and distractor auditory stimuli (6 colour or non-colour words). Each pair was presented three times, making a total of 54 trials for the VT.

#### 2.3.2. Experiment 2: Auditory Task (AT)

The auditory task (AT) was similar, but with an auditory target of colour words, and distractors of either a colour or a non-colour word presented on the screen. The effect of selective attention on the auditory modality was tested with target stimuli from the auditory domain and distractor stimuli from the visual domain. During the training for the task and again before the actual task began, participants were asked to attend to the auditory stimuli while ignoring any visual stimuli on the screen. The set-up and number of trials were similar to VT as described earlier, except that for each trial a randomly selected colour word was presented through the headphones as the auditory target stimulus, while a randomly selected visual stimulus from the pool of colour words (congruent distractor) and non-colour words (incongruent distractor) were presented on the screen.

The task was to identify the colour word presented as the auditory stimulus through the headphones, by clicking the corresponding on-screen button using the mouse.

#### 2.3.3. Post-Test: Task-Load Questionnaire

For measuring the subjective perception of load for visual and auditory tasks, the NASA load TLX questionnaire was distributed to the participants, each receiving one questionnaire after each task. The questionnaire asked the participants to rate the task subjectively on a set of six scales (Mental, Physical, and Temporal Demand; Effort, Frustration, and Performance)on a rating sheet. Each scale was presented as a line divided into 20 equal intervals. The participants marked their responses using tick marks on the given rating scales. Ratings were obtained after each task was completed. Computerised analysis (from NASA Ames Research Centre) was employed to calculate the magnitude of load according to the participant ratings [39].

### 2.4. Measures

Within each task, a trial was considered to be correctly attempted if the participant clicked the button corresponding to the colour word presented as the target (visual in VT, and auditory in AT); otherwise, the trial was deemed incorrectly attempted. Each trial was considered a data point; a score was assigned for each correctly attempted trial, while an incorrect attempt received no score. Total numbers of correct attempts were used for statistical analysis.

For each trial, the response time (RT) in milliseconds was calculated as the time taken from the presentation of the on-screen buttons to the event of the mouse click on one of the buttons. Load scores from NASA load TLX indicated the task load.

### 2.5. Catch Condition

In Experiment 1 (VT), where the auditory distractors supplied to the ear were congruent and incongruent in nature, the gender of the audio inputs were changed exactly 3 times. This change in gender of the audio inputs while performing the visual task was the catch condition.

In Experiment 2 (AT) the visual distractors, both congruent and incongruent, were displayed on the screen. In this case the catch condition was a change in font size from the existing stimuli size to almost double to that of the visual inputs. The change in font size of the visual inputs happened exactly 3 times.

The catch conditions in both the experiments were presented at regular intervals, ensuring that the participant did not encounter the catch condition in back-to-back trials. If the perceptual load of any of the tasks was deemed to be high, it was assumed that the congruent and incongruent distractors and the catch conditions would not be processed.

## 3. Results

The G*Power test was conducted to find the power (1 − β err prob) using an F test—ANOVA: repeated measures within-between interaction. This post hoc analysis was carried out to compute achieved power. The effect size (f) was 0.25 and the α error probability was set at 0.05. The power achieved was (1 − β err prob) = 0.913.

Across the two tasks, we were interested in the effect of modality on perceptual load, and the effect of congruency of the distractor with the target. To this end, the mean accuracy scores and mean RTs were calculated as a function of the effect of congruency of the distractors and type of modality on the performance of the participants.

### 3.1. Accuracy

The plot of mean accuracy scores for AT and VT is shown in Figure 2. The accuracy scores were lower for VT with congruent distractors (M = 0.92, SD = 0.25) compared with incongruent distractors (M = 0.93, SD = 0.24). AT had better accuracy scores for congruent and incongruent distractors, compared with VT (congruent M = 0.99, SD = 0.05; incongruent M = 1.00, SD = 0.00). Furthermore, 2 × 2 ANOVA was conducted to examine the effect of the type of modality (i.e., AT vs. VT) on accuracy scores. The result showed that the modality had a significant effect, F = 111.56; *p* < 0.001. The congruency type did not have a significant effect on the type of modality (F = 0.92, *p* = 0.33).

### 3.2. Response Time

The plot of mean response-time scores for AT and VT is shown in Figure 3. The 2 × 2 logRT ANOVA for type of modality on response time showed a significant effect, F = 17.26, *p* < 0.001; refer to Figure 4 for the distribution of RT data points. The RTs for VT with incongruent distractors were longer (M = 0.99, SD = 2.2) compared with congruent distractors (M = 0.72, SD = 0. 45).

AT in general required shorter RTs (congruent M = 0.69, SD = 0.36; incongruent M = 0.69, SD = 0.35) compared with VT. Variable congruency type had a significant effect on type of modality of tasks (F = 11.19, *p* < 0.001); refer to Table 1 for the ANOVA results.

Figure 5 reports the results from the post hoc test for the type of task modality (AT and VT) and the congruency of distractors. There was a significant interaction between incongruent auditory distractors and VT.

### 3.3. Load

The NASA load TLX was employed to calculate separately the perceived load scores for both visual and auditory tasks. Mean load scores for the types of modality are shown in Figure 6. For the participants who completed VT first, the mean visual load was 43.60, and the mean auditory load was 33.17. For the participants who undertook AT first, the mean auditory load was 36.03, and the mean visual load was 46.69. This shows that irrespective of the order in which the tasks were performed, the mean load of VT was consistently higher than that of AT. Consequently, while the mean load for AT remained the same irrespective of the order in which the tasks were performed, the mean load for VT increased substantially when VT followed AT. The MANOVA results for mean load scores and the order in which participants completed the tasks showed no significant effect of task order on the subjective load scores (F = 2, *p* = 0.937).

Pearson’s correlation coefficient test was conducted to determine whether any correlation existed between load and RT, or between load and accuracy across modalities. The results show that for AT, there was no correlation between load and RT (*r* = 0.122, t = 0.50, *p* = 0.618) or load and accuracy (*r* = −0.19, t = −0.82, *p* = 0.42). However, there was a positive medium correlation for VT between load and RT (*r* = 0.42, t = 1.926, *p* = 0.071), and a negative medium correlation between load and accuracy (*r* = −0.311, t = −1.34, *p* = 0.19).

For the catch conditions across both experiments, a paired t-test was conducted. For the AT, /t/ = 3.5, there was a significant difference between participants observing (no. of catch conditions = ≤3) and not observing the catch conditions (no. of catch conditions = 0). For the VT, /t/ = 1.75, there was no significant difference between participants observing (no. of catch conditions = ≤3) and not observing the catch conditions (no. of catch conditions = 0). Refer to Figure 7 for a summary of catch conditions for each task.

## 4. Discussion

The objective of the current study was to determine whether the congruency and modality of distractors affect perceptual load of tasks. The results indicate that there is indeed had an effect of on the response times and accuracy scores of the participants.

Most of the earlier multimodal studies on attention included targets and congruent distractions of the same modality, with incongruent distractions of a different modality [20,27,29,40]. These studies showed that congruent distractions interfered with targets more than incongruent distractions. When the semanticity of the distractors was the determinant of the congruency, compared with incongruent distractors (with less semantic similarity to the target), congruent distractors (with greater semantic similarity) were shown to have a greater interference effect on the performance of participants.

Our first finding was based on analysing the effect of modality, using congruent and incongruent distractors with a different modality than the target. We alternated between the target and the distractors by switching the modalities from auditory to visual and vice versa. The results show that the auditory distractors interfered more while subjects were performing VT, whereas the visual distractors did not interfere so greatly with AT. The accuracy scores were higher for AT with visual distractions compared with VT with auditory distractors. This finding indicates that modality plays an important role when selectively attending to a particular target. The results explain why certain everyday visual tasks such as driving, where accidents might be caused due to listening to phone conversations, are more prone to interference from auditory distractions. The present study indicates that auditory distractors, especially distractors incongruent to the target, cause higher levels of interference while performing VT.

Our second finding was based on the effect of distractors on the target. While performing a task, studies show that distractors congruent to the target caused more interference compared with incongruent distractors. Previous studies [10,15,18,41] of distractor interference had used responses provoked by congruent or incongruent distractor stimuli alongside the target. The present study eliminated this response–competition paradigm involving the distractors, as the participants were not required to respond to distractors while performing the task. Previous studies of the effects of congruency have generally used only a single modality, i.e., targets and distractors both of the same modality. The present study employed targets and distractors of different modalities, with different congruency ranges, in effect better mimicking a real-world scenario. The results indicate that incongruent auditory distractors were more distracting, with the RT for VT much longer compared with AT. On the contrary, the RT for congruent distractors in both modalities remained almost the same. Our post hoc results also confirm this (Figure 5). This shows that distractors incongruent to the target, irrespective of their modality, interfered with participants’ selective attention and had an effect on their performance.

It should be noted that the congruency between targets and distractors in both AT and VT in our study can be classified as semantic congruency. Previous studies on cross-modal semantic congruency show that multisensory stimuli affect attentional control [36,42]. Our results showed that incongruent distractors were more distracting than congruent distractors, and congruent distractors had no effect on target selection. The latter may be due to the reallocation of attentional resources to the target stimuli facilitating the performance of participants, as we included cross-modal semantic congruency in our tasks [43]. A previous study found that attentional load did not affect the integration of audio–visual stimuli which were semantically congruent to the target, but also revealed potential suppression of the alertness effects induced by incongruent stimuli [42]. We also observed no effect of semantically congruent distractors on RT for AT or VT when there was a shift in attentional load within the tasks. The load of the tasks did not suppress the effect of incongruent stimuli on selective attention. Irrespective of cross-modality, the incongruent semantic distractors were more distracting during the tasks. The extent of interference from incongruent distractors reflected in slower RTs and lower accuracy rates might be dependent on the high working memory load or high cognitive load. High cognitive load induced by the incongruent condition results in greater interference from incongruent distractors.

Our third result relating to the load induced by tasks its effect on distractors stands in contrast to the findings of previous research in the field of PLT [2,29]. PLT suggests that higher load is accompanied by lower distractor interference, and lower load allows higher distractor interference. In the present study, the subjective load measured using the NASA TLX questionnaire indicated higher load scores for VT compared with AT. Participants reported higher load for VT when it was performed after AT. According to previous studies, this should have eliminated the interference effect of congruent as well as incongruent auditory distractors on VT. However, the Pearson’s correlation results for VT load showed a medium positive correlation with RT and a medium negative correlation with accuracy. This indicates that in the high-load task (VT), the RT of the participants increased and there was a drop in accuracy rates. Although VT was marked as a high-load task, it was more affected than AT by distractors. In VT, 24 participants reported noticing the catch condition, compared with only 12 participants noticing it in AT. According to previous studies in PLT [27,29], AT should have shown higher distractor interference, as the participants in our study reported it to be a low-load task.

The Pearson correlation results showed no significant effect of load on RT or accuracy for AT. Contrary to previous findings [27,44,45,46], which suggest that high-load tasks improved performance by effectively blocking distractions, the present study showed comparatively low performance in the high-load VT compared with the low-load AT. Modality, therefore, should be considered a significant parameter when designing tasks in PLT studies.

## 5. Conclusions

The present study establishes that congruency of distractors and targets affects selective attention and the perceptual load of tasks. It also seems that auditory distractors in visual tasks cause more subjective load than visual distractors in auditory task. Previous studies in PLT have indicated that if the load of a task is particularly high, neither the modality nor congruency of distractors should affect the performance of participants. Contrary to that notion, our results indicate that even when the load is high, congruency affects selective attention. Our results suggest that the effects of modality should be considered when designing tasks for the study of selective attention. These results emphasise that modality is as influential as load in terms of its effects on selective attention. In future, further studies should be performed with a larger pool of participants from varying backgrounds to determine the effects of other parameters including culture, gender, and socio-economic strata, to obtain richer results.

## Figures and Tables

**Figure 1 behavsci-13-00051-f001:**
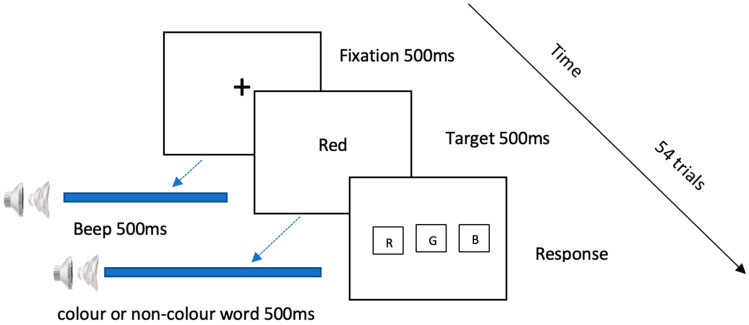
Schematic representation of a trial in the visual task (VT). The visual fixation ‘+’ was presented with a simultaneous auditory fixation beep. The target presented on the screen was a colour word. The distractor, simultaneously presented auditorily, was either a colour or a non-colour word displayed on the screen. The response was recorded via on-screen buttons.

**Figure 2 behavsci-13-00051-f002:**
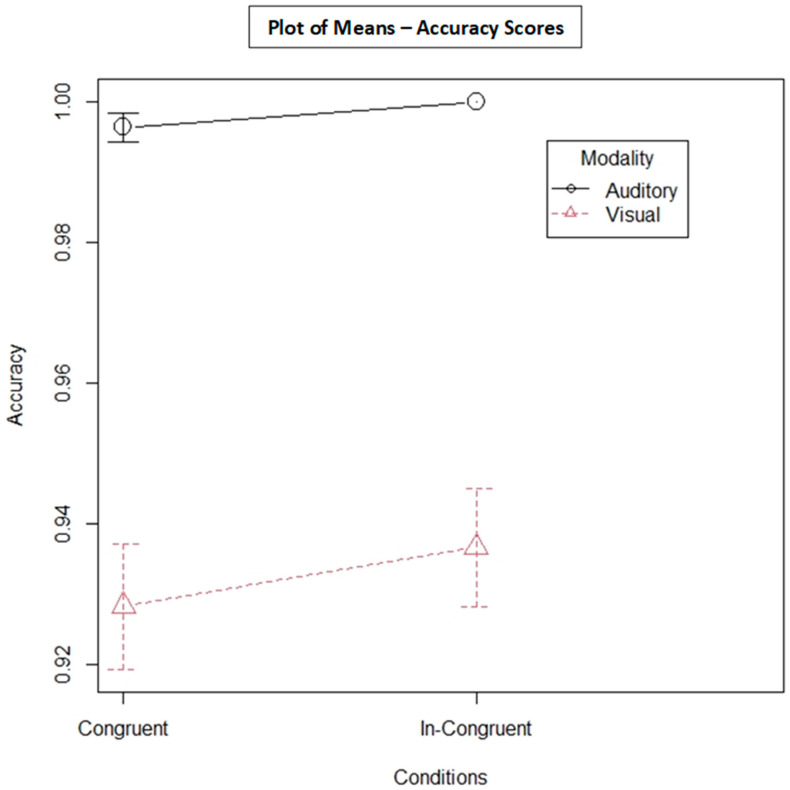
Plot of mean accuracy scores for visual and auditory tasks. Note that the *Y*-axis does not begin at zero.

**Figure 3 behavsci-13-00051-f003:**
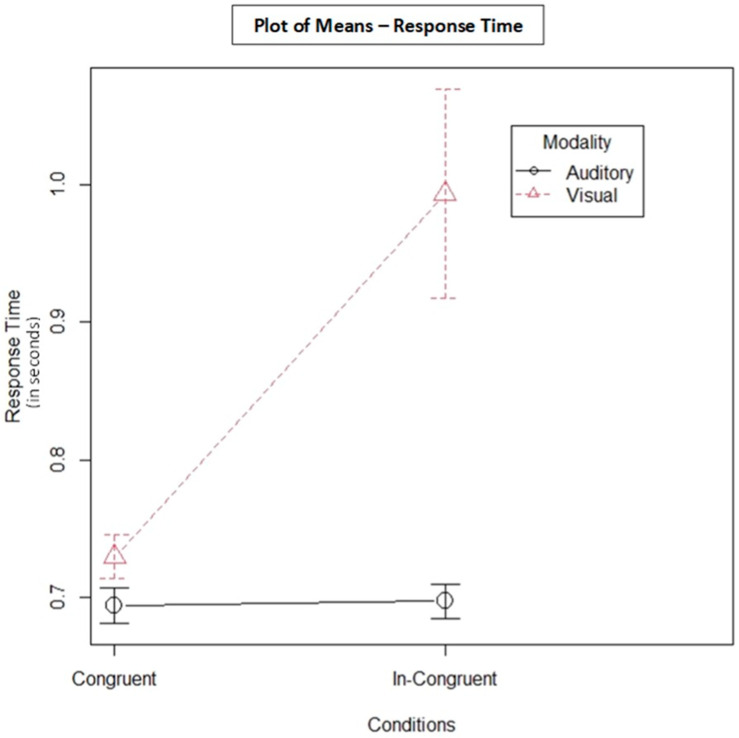
Mean response times for auditory and visual tasks. Note that the *Y*-axis does not begin at zero.

**Figure 4 behavsci-13-00051-f004:**
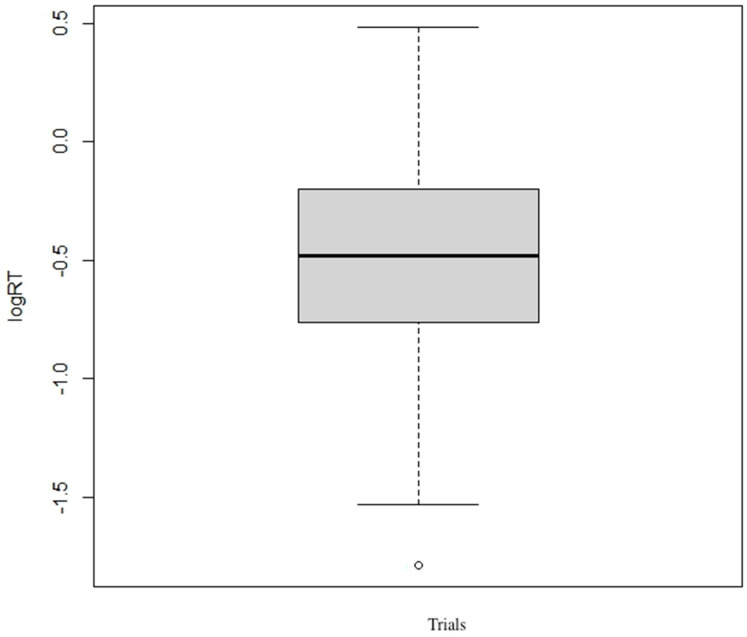
Box plot showing the distribution of response times in seconds from all participants.

**Figure 5 behavsci-13-00051-f005:**
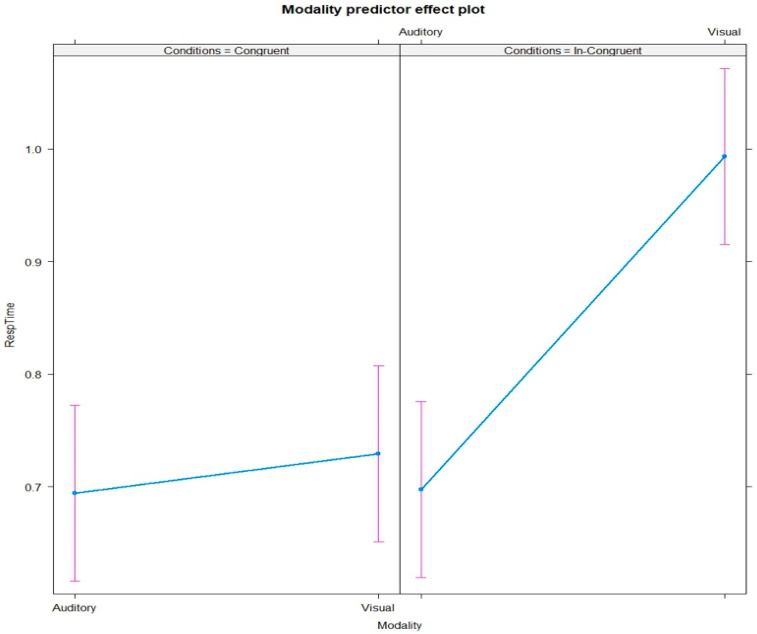
Plot of post hoc modality predictor effect for interaction between condition (congruent and incongruent distractors) and modality (auditory or visual) of the tasks.

**Figure 6 behavsci-13-00051-f006:**
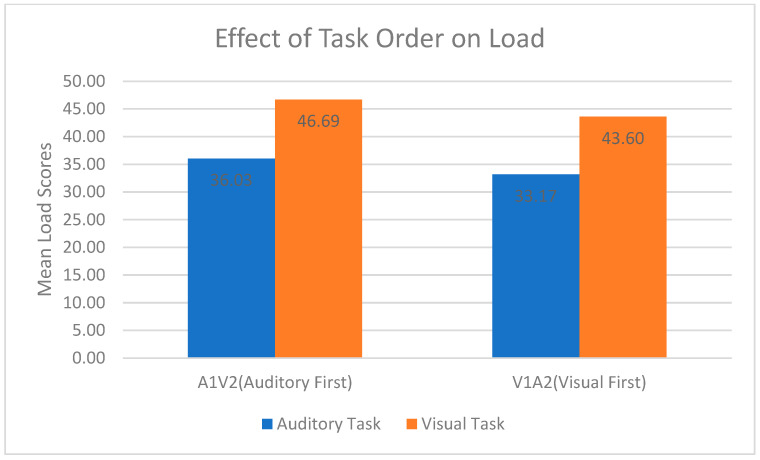
Mean load scores according to type of modality.

**Figure 7 behavsci-13-00051-f007:**
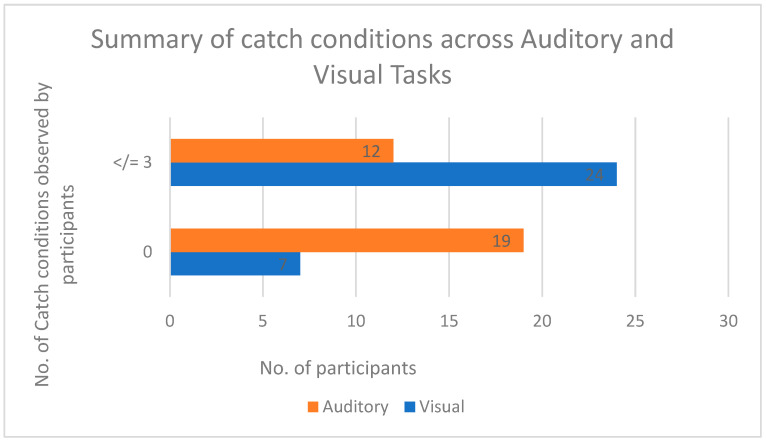
Summary of catch conditions across auditory and visual tasks.

**Table 1 behavsci-13-00051-t001:** Two-way analyses of variance for accuracy and response times in auditory and visual tasks. Significance codes: 0 ‘***’ 0.001 ‘**’ 0.01.

Measures	Sum Sq	F	Pr (>F)
Accuracy			
Congruency	0.030	0.9220	0.3370
Modality	3.614	111.5601	<2 × 10^−16^ ***
Congruency × Modality	0.005	0.1475	0.7009
Response Time			
Congruency	14.9	11.194	0.0008298 ***
Modality	22.9	17.262	0.00003337 ***
Congruency × Modality	14.2	10.710	0.0010762 **

## Data Availability

Data are available at https://osf.io/3xu52/?view_only=fdd6a1e5178b40dbb0feb3b0cff7f612.

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
