# Peer review of "An Exploration of the Effects of Cross-Modal Tasks on Selective Attention"

_behavsci, 2023, doi:10.3390/bs13010051_

Round 1

Reviewer 1 Report

The study examines Perceptual Load Theory (PLT) using a cross-modal task and the subjective assessment of the task load. The results show some interesting results. Overall, I think the manuscript is well-written and the design is clear, and the results are very interesting. However, there are still some issues needed to be addressed.

Lavie also proposed the concept of cognitive load. Although high perceptual load enhances selection, resulting in successful inhibition of distracting stimuli, high cognitive load diminishes the effectiveness of attentional selection, causing failure to inhibit distracting stimuli (Lavie, 2005, 2010).  It seems to me that the subjective assessment here is indicative of cognitive load, instead of perceptual load. I think the authors need to describe the two types of load and clearly articulate why they think they measure “perceptual” load.

Second, as mentioned by the authors, the perceptual load manipulation here is not conventional because it involves semantic processing across two modalities. Thus, I am not sure whether this congruency manipulation can be certainly categorized as a “perceptual” load, or it is actually a cognitive load manipulation. I would predict that congruent condition should be considered as low cognitive load (as should also be reflected by subjective report), resulting in faster RT and higher accuracy.

The subjective load should report all conditions, so the congruent and incongruent conditions in the aud and vis tasks. If the load report indeed measures the perceptual load, then higher perceptual load values should be observed in the congruent condition.

Many statistical comparisons are not reported. For example: is there an interaction in RT (3.2)? 2x2 ANOVA should be performed for 3.3 as well. P values are missing (line 318-323).

Fig. 2 results from catch trials seem to suggest that subjects did not pay attention to do the catch trial task, as the number of subjects should be higher in the upper two boxes than the lower two boxes. I am thus not sure what message can be delivered from this catch trial results.

Line 10 Perceptual Load Theory (PLT)

Line 99-103: Although I agree that it is also good to measure the load directly through the subjective report, I don’t think the definition of the perceptual load is problematic. I think there is a clear operational definition in general to define the level of perceptual load.

Line 143 “Expectation 1: There would be no significant difference in response time and accuracy scores 143 of the participants for tasks from any modality.” I don’t think behavioral results would be the same if you don’t measure the threshold first.

Line 333: missing unit in Y axis. sec?

Author Response

RESPONSE TO REVIEWER

We would like to thank the reviewer for thought-provoking and helpful remarks. The reviewer illuminated the aspects of the manuscript that needed attention. We were able to work on those areas. The reviewer’s remarks are in bold-italics. Our responses are underneath each remark. The changes have been made using track change in the manuscript in various places. 

Lavie also proposed the concept of cognitive load. Although high perceptual load enhances selection, resulting in successful inhibition of distracting stimuli, high cognitive load diminishes the effectiveness of attentional selection, causing failure to inhibit distracting stimuli (Lavie, 2005, 2010).  It seems to me that the subjective assessment here is indicative of cognitive load, instead of perceptual load. I think the authors need to describe the two types of load and clearly articulate why they think they measure “perceptual” load.

Second, as mentioned by the authors, the perceptual load manipulation here is not conventional because it involves semantic processing across two modalities. Thus, I am not sure whether this congruency manipulation can be certainly categorized as a “perceptual” load, or it is actually a cognitive load manipulation. I would predict that congruent condition should be considered as low cognitive load (as should also be reflected by subjective report), resulting in faster RT and higher accuracy.

The subjective load should report all conditions, so the congruent and incongruent conditions in the aud and vis tasks. If the load report indeed measures the perceptual load, then higher perceptual load values should be observed in the congruent condition.

This is a very good point. We commit now in the manuscript the observation made by the reviewer that our study deals with cognitive load as well as perceptual load. Taking care of this point allowed us to better verbalize the motivation of using the kind of stimuli we used. The changes have been made using track change in the manuscript on various places.

Many statistical comparisons are not reported. For example: is there an interaction in RT (3.2)? 2x2 ANOVA should be performed for 3.3 as well. P values are missing (line 318-323).

Now we are reporting more results with graphs. This can be seen in the results section.

Fig. 2 results from catch trials seem to suggest that subjects did not pay attention to do the catch trial task, as the number of subjects should be higher in the upper two boxes than the lower two boxes. I am thus not sure what message can be delivered from this catch trial results.

We have now added the catch trial result analysis in the discussion.

- Line 10 Perceptual Load Theory (PLT)

This has been edited.

Line 99-103: Although I agree that it is also good to measure the load directly through the subjective report, I don’t think the definition of the perceptual load is problematic. I think there is a clear operational definition in general to define the level of perceptual load.

We agree that there is a clear definition of perceptual load. However, the problem that we are trying to point out is of the categorization of tasks being high- or low- perceptual load, and whether this categorization is made by the experimenter before the experiment, or if the categorization is made after observing the distraction interference result during the experiment.

Line 143 “Expectation 1: There would be no significant difference in response time and accuracy scores 143 of the participants for tasks from any modality.” I don’t think behavioral results would be the same if you don’t measure the threshold first.

We are stating the naïve hypothesis here which is that perceptual load on selective attention shall be agnostic to which modality is engaged in performing the task and which modality distracting.

Line 333: missing unit in Y axis. sec?

This has been changed.

Reviewer 2 Report

In the manuscript, the authors reported interestingly and extended data on the evaluation of the effect of cross-modality (target and distractor from separate modalities) and congruency (similarity of target and distractor) on selective attention,   showing  an interesting  effect of modality (congruency of the distractors) and no effect of load on selective attention.  The study was well structured, even if the population reported is based on a small number of participants, thus further studies are advisable in future.

Author Response

In the manuscript, the authors reported interestingly and extended data on the evaluation of the effect of cross-modality (target and distractor from separate modalities) and congruency (similarity of target and distractor) on selective attention,   showing  an interesting  effect of modality (congruency of the distractors) and no effect of load on selective attention.  The study was well structured, even if the population reported is based on a small number of participants, thus further studies are advisable in future.

We would like to thank the reviewer for the observations. We agree with the suggestion that outlook of the study should spell out the fact that the study can be improved with larger number of participants and taking more parameters into consideration. This has been added in the manuscript. The changes have been made using track change in the manuscript.

Reviewer 3 Report

Nambiar and Bhargava intend to offer a new focus on perceptual load theory by studying the effect of modality and distractor congruency on a perceptual task, and using self-assessed measures of load. My main concern is that the statistical testing of the hypothesis is not properly done. Many of the comparisons are just comparisons between average values, but some main or posthoc effects or interactions are not tested or reported appropriately. For example:

"While the mean load for AT remained the same irrespective of the order in which AT were performed, the mean load for VT increased substantially when VT followed AT." -> Has this been statistically assessed?

"The results  show that for AT, there is no correlation between load and RT (r = 0.122) or load and accuracy (r = -0.19)." -> Are these correlations different from zero, or from each other?

Also, presenting the whole distribution of data points in the plots rather than just means and standard deviations would help the reader get an idea of the actual effects.

I would encourage the authors to improve the reporting of effects as to better support their claims and conclusions.

Author Response

We found the reviewer's comments to be constructive and very helpful with respect to improving the statistical reporting. The reviewer’s remarks are in bold-italics. Our responses are underneath each remark. The changes have been made using track change in the manuscript in various places.

Nambiar and Bhargava intend to offer a new focus on perceptual load theory by studying the effect of modality and distractor congruency on a perceptual task, and using self-assessed measures of load. My main concern is that the statistical testing of the hypothesis is not properly done. Many of the comparisons are just comparisons between average values, but some main or posthoc effects or interactions are not tested or reported appropriately. For example:

"While the mean load for AT remained the same irrespective of the order in which AT were performed, the mean load for VT increased substantially when VT followed AT." -> Has this been statistically assessed?

We have added the MANOVA results for the subjective load assessment to the manuscript. We have also conducted the posthoc effect test and have reported the results.

"The results  show that for AT, there is no correlation between load and RT (r = 0.122) or load and accuracy (r = -0.19)." -> Are these correlations different from zero, or from each other?

We have edited this.

Also, presenting the whole distribution of data points in the plots rather than just means and standard deviations would help the reader get an idea of the actual effects. I would encourage the authors to improve the reporting of effects as to better support their claims and conclusions.

We have added the plot with spread of our data points to the manuscript.

Round 2

Reviewer 1 Report

The author's have addressed my questions.